# Vesicles Shed by Pathological Murine Adipocytes Spread Pathology: Characterization and Functional Role of Insulin Resistant/Hypertrophied Adiposomes

**DOI:** 10.3390/ijms21062252

**Published:** 2020-03-24

**Authors:** Camino Tamara, Lago-Baameiro Nerea, Bravo Susana Belén, Sueiro Aurelio, Couto Iván, Santos Fernando, Baltar Javier, Casanueva F. Felipe, Pardo María

**Affiliations:** 1Grupo Obesidómica, Área de Endocrinología, Instituto de Investigación Sanitaria de Santiago de Compostela (IDIS), Xerencia de Xestión Integrada de Santiago (XXIS/SERGAS), 15706 Santiago de Compostela, Spain; tamara_cm_10294@hotmail.es (C.T.); neelago.18@gmail.com (L.-B.N.); ivan_couto@hotmail.com (C.I.); ffsantosbenito@gmail.com (S.F.); javier.baltar.boileve@sergas.es (B.J.); 2Unidad de Proteómica, Instituto de Investigación Sanitaria de Santiago (IDIS), Xerencia de Xestión Integrada de Santiago (XXIS/SERGAS), 15706 Santiago de Compostela, Spain; sbbravo@gmail.com; 3Grupo Endocrinología Molecular y Celular, Instituto de Investigación Sanitaria de Santiago (IDIS), Xerencia de Xestión Integrada de Santiago (XXIS/SERGAS), 15706 Santiago de Compostela, Spain; aurelio.manuel.martis.sueiro@sergas.es (S.A.); felipe.casanueva@usc.es (C.F.F.); 4CIBER Fisiopatología Obesidad y Nutrición, Instituto de Salud Carlos III, 15706 Santiago de Compostela, Spain; 5Servicio de Cirugía Plástica y Reparadora, Xerencia de Xestión Integrada de Santiago (XXIS/SERGAS), 15706 Santiago de Compostela, Spain; 6Servicio de Cirugía General, Xerencia de Xestión Integrada de Santiago (XXIS/SERGAS), 15706 Santiago de Compostela, Spain

**Keywords:** extracellular vesicles, adipocytes, insulin resistance, lipid hypertrophy, inflammation, proteomics, SWATH

## Abstract

Extracellular vesicles (EVs) have recently emerged as a relevant way of cell to cell communication, and its analysis has become an indirect approach to assess the cell/tissue of origin status. However, the knowledge about their nature and role on metabolic diseases is still very scarce. We have established an insulin resistant (IR) and two lipid (palmitic/oleic) hypertrophied adipocyte cell models to isolate EVs to perform a protein cargo qualitative and quantitative Sequential Window Acquisition of All Theoretical Mass Spectra (SWATH) analysis by mass spectrometry. Our results show a high proportion of obesity and IR-related proteins in pathological EVs; thus, we propose a panel of potential obese adipose tissue EV-biomarkers. Among those, lipid hypertrophied vesicles are characterized by ceruloplasmin, mimecan, and perilipin 1 adipokines, and those from the IR by the striking presence of the adiposity and IR related transforming growth factor-beta-induced protein ig-h3 (TFGBI). Interestingly, functional assays show that IR and hypertrophied adipocytes induce differentiation/hypertrophy and IR in healthy adipocytes through secreted EVs. Finally, we demonstrate that lipid atrophied adipocytes shed EVs promote macrophage inflammation by stimulating IL-6 and TNFα expression. Thus, we conclude that pathological adipocytes release vesicles containing representative protein cargo of the cell of origin that are able to induce metabolic alterations on healthy cells probably exacerbating the disease once established.

## 1. Introduction

Extracellular vesicles (EVs), and exosomes in particular, have recently arisen as a very specialized mechanism for cell to cell communication at a local and distant level [1]. EVs comprise spherical vesicles of diverse size (30 to 1000 nm) and origin released in effect by all cell types; thus EVs are found in the majority of biological fluids, and also in the cell culture medium of most cell types [2]. Depending on the size and biogenesis, EVs comprise: microvesicles (100 nm–1 µm) originated by blebbing of the plasma membrane, exosomes (30–100 nm) assembled in multivesicular endosomes (MVEs) that are secreted by exocytosis, and larger vesicles (50–5000 nm) that include apoptotic bodies released by cells prior to apoptosis [3,4]. Exosomes, microvesicles, and other EVs enclose membrane and cytosolic components such as proteins, lipids, and RNAs; interestingly, this composition is conditioned by the site of biogenesis [5]. 

In particular, exosomes have been recently rediscovered due to their role as a sophisticated intercellular communicating system which implies its fusion or internalization with hosting cells to liberate in the interior its content [3]. Upon interaction with target cells, exosomes affect functionally the recipient cells, inducing them to actively contribute in different physiological and pathological processes [6]. Moreover, since the molecular cargo of the exosomes mirrors physiological and pathophysiological changes of the cell or tissue of origin, the exosomes have arisen as potential biomarkers for the diagnosis of the disease [7]. These characteristics have generated great expectations around EVs and exosomes in particular, making the study of these vesicles a hot topic in many fields of research.

The role of exosomes on cell communication in the development of metabolic disease is still poorly known. There is increasing evidence implicating EVs in obesity-associated metabolic deregulation [8], and more precisely, in the local and systemic inflammation linked to liver and adipose tissue [9]. Thus, EVs shed by adipose tissue have been proposed to be involved in adipocyte/macrophage cross talk [10,11], and to affect insulin signaling and expression in muscle and liver cells leading to metabolic disease [12,13,14,15]. The biogenesis of EVs, and exosomes in particular, is a dynamic process that correlates with the physiological and pathological status of the parent cell; therefore, vesicles cargo including proteins and miRNAs varies accordingly. Our hypothesis is that EVs liberated by adipocytes vary along the differentiation process and during the development of hypertrophy and insulin resistance probably exacerbating adipose tissue alteration in obesity. 

In this manuscript, we isolate and characterize vesicles shed by a murine model of adipocyte differentiation during lipid atrophy and insulin resistance. Differences among normal and hypertrophied/insulin resistant adipocyte-vesicles cargo are identified and characterized. Moreover, the functional effect of vesicles shed from pathological hypertrophied and insulin resistant adipocytes on healthy adipocytes and macrophages is assessed. 

## 2. Results

### 2.1. Adipocytes Shed Extracellular Vesicles That Change Their Protein Cargo with Lipid Atrophy and High Glucose/High Insulin (HGHI)-Promoted Insulin Resistance 

Cell models of adipocyte lipid atrophy by oleic acid and palmitate, and a model of insulin resistance by high glucose-high insulin (HGHI) treatment were established. Exposure of C3H10T1/2 adipocytes to palmitate (500 µM) and oleic acid (1 mM) significantly increased lipid accumulation as assed by oil red staining, while no changes were observed on those adipocytes exposed to high glucose and high insulin treatment (4.5 g/L glucose; 5 µg/mL insulin; 24 h) (Appendix A). Moreover, both lipid atrophy and HGHI exposure induced insulin resistance as demonstrated by a significant reduction on P-Akt/Akt signaling in these cells after 10 min stimulation with insulin (Appendix A). In an attempt to elucidate if these metabolic changes are reflected in the liberated extracellular vesicles, secretomes from the three cell models and from control differentiated adipocytes were collected and processed for extracellular vesicle isolation (Figure 1A). An initial nanoparticle tracking analysis (NTA) of the resulting pellets showed the identification of vesicles with an average size of 189.3 (+/−1.7) nm for control cells, 144.9 (+/−5.4) nm for palmitate, 145.0 (+/−8.9) nm for oleic acid, and 129.1 (+/−4.3) nm for those isolated from HGHI exposed cells (Figure 1B–F). Interestingly, hypertrophied cells with palmitate and oleic acid showed less particle concentration per frame but double protein content compared to vesicles from control and HGHI cells (Figure 1F).

To obtain information about vesicle protein composition and cargo, and its variation depending on the metabolic/pathological status of the cell of origin, isolated vesicles were characterized by proteomics qualitative (DDA) and quantitative (SWATH) mass spectrometry analysis (n = 4 independent experiments). Thus, the qualitative proteome analysis considering those proteins present in 3 out of 4 independent experiments showed a striking difference between vesicles isolated from pre and differentiated adipocytes, identifying 138 different proteins that increased up to 825 respectively (Appendix A). We compared our data with previous proteomics reports of 3T3 murine adipocyte cells-derived EVs to validate our cell model, finding that among those common proteins between pre- and mature C3H10T1/2 EVs, 98% were previously described in vesicles from differentiated 3T3-L1 cells [16] and 60% from 3T3-F442A [17] (Appendix A). However, our study reveals many new proteins on EVs secreted by mature C3H10T1/2 adipocytes; thus, from those proteins exclusively detected in differentiated C3H10T1/2, only 36% were previously described in 3T3-L1 EVs, and 21% in 3T3-F442A (Appendix A). Additionally, 577 proteins were identified in EVs from hypertrophied adipocytes with palmitate, 291 in those vesicles shed by adipocytes treated with oleic acid, and 588 from insulin resistant adipocytes (HGHI) (Figure 2A; Appendix A). EVs from control mature adipocytes and those isolated from pathological shared 229 common proteins (Figure 2A, B; Appendix A); however, 209, 67, 32, and 5 different proteins were exclusively present in EVs from adipocytes, HGHI, palmitate, and oleic acid, respectively (Figure 2A; Appendix A). A list of all identified proteins on EVs from the three pathological cell models was compared to previous reports showing normal healthy 3T3 differentiated adipocytes-shed EVs, human primary SAT adipocytes EVs, and to the white adipose tissue gene enriched database [18] (Appendix A). Additionally, we compared identified proteins to our previous study analyzing EVs from human obese whole visceral and subcutaneous adipose tissues [19], finding several common proteins (Appendix A). 

The functional classification of analyzed EVs protein cargo showed as expected that the majority of the identified proteins were present in the Vesiclepedia (Figure 2B). Moreover, it was interesting to see that EVs from differentiated adipocytes carried, compared to preadipocytes, a higher percentage of proteins linked to protein metabolism, general metabolism, and energy pathways, and a decrease on cell growth and/or maintenance proteins (Figure 2C). On the other hand, EVs isolated from insulin resistant adipocytes (HGHI) showed proteins implicated in diabetes pathways, that were not present on EVs secreted by lipid hypertrophied cells (Figure 2D). EVs shed by adipocytes treated with palmitate and oleic acid contained proteins implicated in energy pathways, general metabolism, cell growth, protein metabolism, and signal transduction that were increased in oleic compared to palmitate; however, those vesicles isolated from palmitate treated adipocytes, contained signal transduction and cell communication proteins that were no present in those treated with oleic acid (Figure 2E). Interestingly, certain identified proteins exclusively found in the three pathological cell models were present within the obesity and diabetes gene card (GeneCards, The human gene database https://www.genecards.org).

The quantitative label-free mass spectrometry analysis by differential Sequential Window Acquisition of all Theoretical fragment-ion spectra (SWATH) showed those proteins elevated and decreased in EVs from differentiated adipocytes compared to non differentiated cells, and also comparing each cell model to control adipocytes (*p* ≤ 0.05, and a fold change ≥ 1.5) (Figure 3A). The list of differences among groups with a fold change ≥2 is shown in Appendix A and comparison with those proteins elevated in human obese visceral (VAT) and subcutaneous (SAT) adipose tissue EVs [19] in Appendix A. Principal component analysis (PCA) of the obtained results shows a clear separation of EVs liberated by those adipocytes treated with palmitate and oleic acid; on the other hand, those treated with high glucose and insulin were closer to control differentiated adipocytes (Figure 3B). SWATH cluster plot analyses of EVs proteins identified in adipocytes compared to EVs from mature adipocytes, HGHI, palmitate and oleic acid are shown in Appendix A. The comparison of elevated and decreased proteins in vesicles from the three pathological cell models compared to control differentiated vesicles, including those common to the three metabolic insults, are listed (Figure 3C,D; Appendix A); protein cargo of HGHI, palmitate and oleic EVs were also compared among each other (Appendix A). A summary with representative proteome maps of protein cargo from vesicles secreted by control differentiated adipocytes, and those upregulated or decreased in the pathological adipocytes compared to EVs from control differentiated cells were represented (Figure 3E,F). 

To confirm the proteomics results, we validated vesicle markers [programmed cell death 6-interating protein PDC6I (Alix), Tumor susceptibility gene 101 protein (TSG101), Syntenin 1, and CD81, including previously suggested adipose tissue-specific EV marker, perilipin 1, by immunoblot (Figure 3G). Moreover, we assayed proteins of interest and candidate obese biomarkers recently described by us in human obese VAT and SAT that were also identified in the pathological cell models such as ceruloplasmin, mimecan and transforming growth factor-beta-induced protein ig-h3 (TFGBI) (Figure 3G,I). Interestingly, we observed that EV markers syntenin 1, TSG101, and Alix vary with metabolic insults. Additionally, we observed that perilipin 1 and mimecam were upregulated in EVs from lipid hypertrophied adipocytes (Figure 3G,J,K). Confirming the SWATH analysis, we show a striking elevation of TFGBI exclusively on those vesicles liberated by insulin resistant adipocytes (HGHI) compared to control, and especially in relation to hypertrophied vesicles (Figure 3G,I). On the other hand, ceruloplasmin was found elevated in those vesicles from palmitic and especially oleic acid treated adipocytes compared to HGHI or control adipocytes (Figure 3G,H). A list of protein-EV biomarkers candidates from the obtained results based on proteomics analysis, inmunoblot validation, and bibliography, is shown in Table 1. 

### 2.2. Vesicles Shed by Hypertrophied and Insulin Resistant Adipocytes Induce Adipocyte Differentiation

Functional assays were designed to asses if EVs shed by adipocytes participate or modulate the differentiation of neighbor cells, and also to discern if this effect, if any, is altered by pathological vesicle composition. Thus, control normal adipocytes were differentiated in the presence of vesicles isolated from lipid atrophied (palmitate/oleic acid) and HGHI adipocyte cells models. Real time monitoring of adipocyte differentiation showed that oleic acid, palmitate, and HGHI EVs alter the differentiation of control cells compared to those without these vesicles (Figure 4A–C). Thus, we observed a reduction of the cell index on those cells differentiated in the presence of the three types of pathological EVs; becoming statistically significant from differentiation day 2 for palmitate and HGHI, and at day 4 in the three experimental settings (Figure 4D–F).

### 2.3. Vesicles Shed by Hypertrophied and Insulin Resistant Adipocytes Induce Insulin Resistance in Healthy Cells

The treatment of healthy control differentiated adipocytes with vesicles isolated from pathological adipocytes inhibited the insulin pathway stimulation (10 min, 100 mM insulin) as assessed by Akt phosphorylation (Ser 473). Thus, we observed a significant decrease on Akt phosphorylation on healthy adipocytes incubated for 24h with pathological vesicles followed with insulin 100 nM stimulation during 10 min (Figure 5A–C). This effect was especially intense for HGHI vesicles paralleling the insulin resistance of the cells of origin (Figure 5C; Appendix A). Moreover, we observed that combined pathological cell models performing hypertrophy (palmitate/oleic) followed by insulin and glucose treatment (HGHI) exacerbated the deleterious effect; thus, vesicles from combined cell models were even more effective inducing insulin resistance (Figure 5D, E). However, EVs isolated from murine inflamed macrophages (LPS 1 µg/mL) did not affect insulin pathway through Akt (Figure 5F).

### 2.4. Vesicles Shed by Hypertrophied Adipocytes Induce Macrophage Inflammation

Functional assays exposing murine macrophages with lipid hypertrophied vesicles at different concentrations showed a significant increase of inflammation as measured by TNFα (Figure 6A–D) and IL-6 (Figure 6E,F). Interestingly, while a physiological dose of PALM EVs was enough to induce significant macrophage inflammation, it was necessary a double amount of OLEIC EVs to obtain a similar effect on macrophages. Inflammation was not observed when treating macrophages with vesicles from control healthy adipocytes, neither with those isolated from HGHI cells.

## 3. Discussion

In the present manuscript we show for the first time that the protein content of extracellular vesicles (EVs) shed by mature adipocytes is modified and altered by metabolic insults as lipid hypertrophy and insulin resistance (IR). Interestingly, various identified proteins in these EVs were previously described as obesity and IR-related, and also found earlier by us in vesicles liberated by human obese visceral and subcutaneous adipose tissue explants; thus, we propose a panel of potential obese adipose tissue EV-biomarkers. Among then, we show that EVs isolated from palmitate and oleic acid treated adipocytes were characterized by an elevation of ceruloplasmin, mimecan and perilipin 1 adipokines, and vesicles from the IR cell model showed an exclusively striking presence of the adiposity and IR related transforming growth factor-beta-induced protein ig-h3 (TFGBI). Moreover, we show a functional role of EVs secreted by hypertrophied and IR cells as they promote the pathology of the cell of origin to normal healthy adipocytes as assessed by real time hypertrophy assays and insulin pathway. Additionally, we show that EVs shed by lipid atrophied adipocytes promote macrophage inflammation. 

Despite the recent up surge of extracellular vesicles revealing a whole new paradigm on cell to cell communication, the knowledge about the role of these vesicles in metabolic homeostasis is scarce. There is no clear view on the EVs released by metabolism implicated tissues, neither their alteration by pathological conditions [9]. However, recent research has begun to shed light in this matter by describing circulating EVs as metabolic biomarkers for lipid and glucose metabolism in humans [35], or studying vesicles liberated by human and murine adipose tissue explants and cell lines [12,19]. Thus, EVs liberated by adipose tissue have been shown to participate in macrophage-induced insulin resistance [11], promote AT resident macrophages inflammation [10], alter insulin signaling in adipocytes, liver, and muscle cells [15,36], and deregulating the transforming growth factor beta pathway in cultured HepG2 cells [13]. Moreover, it was recently shown that murine adipocyte-derived EVs can regulate POMC expression through the hypothalamic mTOR signaling in vivo and in vitro affecting body energy intake [37]. 

The present study follows our previous work showing, for the first time, the characterization of EVs shed by visceral and subcutaneous human obese and lean adipose tissue explants, revealing depot and obesity candidate biomarkers [19]. In the present work, we have performed the first complete proteome characterization of vesicles isolated from differentiated adipocytes to be compared with the same cells submitted to lipid hypertrophy with palmitate and oleic acid, and also to insulin resistance induced by high glucose and high insulin (HGHI) treatment as previously established and described [38,39]. EVs were isolated, after proving lipid accumulation and a diminution of P-Akt on insulin pathway after insulin stimulation, by serial centrifugation following our previous approach based in the MISEV recommendations [40]. Intriguingly, we observed that lipid hypertrophied adipocytes show a diminution on particle concentration secreted to the cell culture medium compared to control cells as measured by NTA. This was unexpected as it was shown that the presence of stress or pathology including obesity induces an increase on circulating vesicles [41,42]. On the contrary, those IR adipocytes generated by HGHI showed more elevated particles per frame compared to differentiated control cells or to hypertrophied adipocytes. Measuring protein concentration of vesicles pellets from equal starting number of cultured cells showed that palmitate and oleic acid treated cells showed almost the double of protein than control and HGHI vesicles, which may indicate differences in the amount of protein cargo for each metabolic situation. In the same way, prior characterization studies of the murine 3T3-L1 cell line-shed EVs, has shown a diminution on the amount of particles secretion measured by NTA from day 0 to 15 of adipocyte differentiation [43]. In summary, our results show that the concentration of particles obtained by NTA analysis does not correlate with the amount of proteins quantified after lysis of the EVs pellet, neither with the number of different identified proteins by mass spectrometry. In fact, mass spectrometry analysis of isolated EVs in our study show that preadipocytes and mature adipocytes share CD81, ALIX, Syntenin-1, and TSG101; however, exosomal CD9 and CD63 were identified only in vesicles from mature adipocytes, suggesting that the populations and kinetics of secreted vesicles are dynamic and are also modulated by the cell of origin physiology.

To our knowledge, this is the first study analyzing the C3H10T1/2 adipocytes-derived EVs; moreover, the present manuscript shows novel results by identifying protein cargo changes on EVs from the same cells after lipid hypertrophy or under insulin resistance by qualitative and quantitative (SWATH) label free mass spectrometry analysis. Previous EVs proteomics analysis were published using 3T3-L1 and 3T3-F442A adipocyte murine cell lines [16,17], and also from human primary SAT adipocytes in culture of two lean to moderate obese patients [44] whose results were indeed compared to ours herein; however, none of the previous proteomics analysis described vesicle protein cargo under metabolic insults as far we are aware. 

Qualitative proteomics analysis of EVs isolated from the five experimental situations showed a striking difference in the amount of identified proteins. Thus, the number of different proteins identified in EVs from differentiated adipocytes increased six times compared to preadipocytes which might make sense due to the drastic changes promoted by the differentiation cocktail. Additionally, differences on the number of identified proteins by mass spectrometry in secreted EVs vary according to the treatment in palmitate, oleic acid, and HGHI cells. 

It is of interest to highlight the identification of known adipocyte associated proteins and adipokines on the EVs of differentiated cells compared to undifferentiated such as adiponectin, caveolae-associated protein 1, fatty acid synthase, endoplasmin-Grp94, Grp75, caveolin 1/2, adipocyte enhancer-binding protein 1, adipocyte plasma membrane-associated protein, acetoacetyl-CoA synthetase, perilipin1/4, lipoprotein lipase, major vault protein, macrophage migration inhibitory factor (MIF), fatty acid-binding protein 4 and 5 FABP-4/5, glycerol-3-phosphate dehydrogenase, hormone sensitive lipase, chemerin, and Glut1, among others, showing that shed EVs become a good representation of the cell of origin including its metabolic status. More importantly, EVs from hypertrophied and IR adipocytes, showed to carry proteins previously described as obesity and IR-related. From those, 25 different proteins were found by us earlier in vesicles liberated by human obese visceral and subcutaneous adipose tissue explants (32% of proteins were present in human obese VAT vesicles, and 24% in obese SAT [19]. Among those is the ER stress-related protein calreticulin, which regulates adipogenesis, and it was shown to be increased in IR and obesity [45] or S100A6 that belongs to a large family of Ca2+ binding proteins implicated in numerous human diseases, such as rheumatoid disease, acute inflammatory lesions, cardiomyopathies, and cancer [46]. Similarly, we found mimecan protein on IR vesicles and also by immunoblot in EVs from hypertrophied cells paralleling its upregulation in vesicles isolated from human obese visceral adipose tissue [19]. This adipokine is abundantly expressed in AT and considered a satiety hormone that inhibits food intake independent of leptin signaling by inducing IL-1B and IL-6 expression in the hypothalamus [27]. Although it has not been described, we hypothesize that as occurs with leptin, this adipokine may be increasingly secreted during the development of obesity, and probably part of this secretion occurs through EVs as validated by immunoblot in the present research. Other interesting proteins, related to obesity, were also identified in the pathological vesicles of our analysis such as protein/nucleic acid deglycase DJ-1, involved in adipogenesis and obesity-induced inflammation [47], PPIB, an adipogenic factor implicated in obesity [30], or tenascin, involved in obesity via visceral adipose tissue inflammation representing a link with extracellular matrix (ECM) remodeling [48].

In addition, the quantitative analysis of vesicle cargo proteins revealed how proteins of interest vary their amount specifically depending on the metabolic insult, suggesting its utility as vesicle biomarkers of pathology. This is the case for ceruloplasmin, identified initially by the qualitative analysis in lipid hypertrophied vesicles. This protein has been recently described as a novel adipokine overexpressed in AT of obese subjects and in obesity-associated cancer cells [23]; additionally, it was postulated as a biomarker for obesity [24]. Interestingly, our quantitative analysis showed that ceruloplasmin was detected with mild elevation in vesicles from IR adipocytes (2.70 times) that becomes strikingly high on those EVs secreted by oleic (70 times) and palmitate (52 times) hypertrophied cells compared to those shed by control adipocytes. Accordingly, this protein was previously detected by us in EVs from visceral adipose tissue explants of obese patients [19]. Other potential obesity biomarkers were found among those proteins elevated in vesicles from lipid hypertrophied cells such as inter-alpha-trypsin inhibitor heavy chain H3 (ITIH3), Glut4, cathepsin B, osteopontin, CD36, or tissue factor. ITIH3 was the protein with higher fold change in EVs from cells hypertrophied with palmitate (126 times) and oleic acid (203 times) compared to those shed by control adipocytes. This protein belongs to the ITH family of proteins that are strongly associated to inflammation [26]. On the other hand, we showed the elevated presence of Glut4 on hypertrophied vesicles; we might speculate that secretion of this glucose transporter through EVs might be a way of participating in insulin resistance; however, this protein was not identified in those vesicles from the HGHI cell model, thus, this will require further analysis. In the case of cathepsin B, it was reported the secretion of this cysteine peptidase protein during adipocyte hypertrophy in the development of obesity promoting an excessive increase of autophagy, inflammation and macrophage infiltration contributing to metabolic syndrome [21]. Another protein of interest is osteopontin, as it is highly up-regulated in AT in human and murine obesity, and has been functionally involved in the pathogenesis of obesity-induced adipose tissue inflammation and insulin resistance [29]. Our study detected that this protein is 12 times in palmitate and 4 times in oleic EVs compared to control adipocytes; thus we show that not only adipose tissue macrophages secreted this protein, but adipocytes themselves are able to liberate this inflammatory cytokine through EVs. In the same direction is CD36 antigen, which was shown to generate an inflammatory paracrine loop between adipocytes and macrophages facilitating chronic inflammation and contributing to IR [22]. Moreover, tissue factor protein should be mentioned, as inflammation and metabolic changes during the development of obesity was found to increase this protein in adipocytes and macrophages in obese adipose tissue [34,49].

Our study additionally revealed those proteins that differentiate EVs among different pathological adipocyte models bringing out those proteins elevated or exclusive for each condition. This is the situation for 72kDa type IV collagenase (MMP-2), fatty acid-binding protein 4 (FABP-4), and transforming growth factor beta induced protein ig-h3 (TFGBI) that were elevated in EVs from insulin resistant adipocytes by HGHI. MMP-2 was described as elevated at circulating level in obese humans [20], and furthermore, it has been implicated in leptin resistance causing IR and obesity in mice by leptin receptor cleavage [50]. Additionally, the adipocyte protein FABP4 has been previously associated to obesity and metabolic syndrome [25]. However, the most remarkable finding was to find TFGBI strikingly elevated in IR vesicles with almost no signal in vesicles from lipid hypertrophied cells. TFGBI is a protein inducible by TFGB1 that is secreted by many cells, and was shown to bind collagen forming part of the extracellular matrix and able to interact with integrins of the cell surface [51]. It was previously described by us within the top 5 of more highly secreted protein from human obese VAT explants [52]; but more outstandingly, it has been described as a diabetes-risk gene based on mouse and human genetic studies [32,33]. Taking into account that we previously found these three proteins, MMP-2, FABP4, and TFGBI, as elevated in EVs shed from human obese visceral AT compared to subcutaneous, and that visceral AT is considered more insulin resistant [53], they might be good candidates as disease vesicle biomarkers.

On the contrary, other proteins characterized lipid hypertrophied vesicles. This is the case for perilipin 1, identified as a potential novel biomarker to detect adipocyte-derived EVs in circulation [54]. Our study by immunoblot shows that this protein is especially useful to detect EVs from lipid atrophied adipocytes; thus our results are coherent with previous reports describing that perilipin1-EVs are significantly increased in mice with diet-induced obesity (DIO) and in obese humans with metabolic syndrome compared to lean controls [35]. Elevated exclusively in palmitate vesicles is syntaxin-8 whose expression was described previously as increased in VAT of obese patients with type 2 diabetes and related to markers of IR and inflammation [31]. Elevated in vesicles from oleic acid hypertrophied adipocytes are other proteins of interest in relation to adipose tissue and obesity, such as PARK7/DJ-1 and MMP-14. DJ-1 protein is implicated in oxidative stress regulation among other functions, but has been also described as participating in adipogenesis and in obesity-induced inflammation [47]. Furthermore, MMP-14 was found to be overexpressed in obese adipose participating in abnormal lipid metabolism and insulin resistance [28]. Finally, it is also interesting to mention those proteins whose presence was elevated in EVs from non-differentiated adipocytes compared to those from mature cells, which may parallel those mesenchymal stem cells (MSCs) embedded in adipose tissue. Thus, we detected that pentraxin-3 protein is seven times elevated in MSCs vesicles compared to those from the same cells after differentiation. This protein has a protective role in LPS and high fat diet-induced inflammation in murine adipose tissue [55], thus, it may suggest a functional anti-inflammatory role of those vesicles secreted by MSCs as previously described [56].

Apart from the description of the protein cargo of EVs isolated from the different cell models, we performed experiments to assess the role of these vesicles in paracrine cross talk between adipocytes and other adipose tissue resident cells. Thus, we describe for the first time that EVs from hypertrophied and IR C3H10T1/2 cells induce adipocyte differentiation as observed with a decrease on cell index. Both the real-time monitoring and the diminution of the cell index indicating cell differentiation are well-established and documented systems [57]. To our knowledge, this is the first time showing that EVs shed by lipid hypertrophied and IR adipocytes accelerates healthy adipocytes differentiation and hypertrophy. Furthermore, we demonstrate that these same vesicles promote IR by inhibiting the insulin signaling pathway individually, and especially when adding first vesicles from hypertrophied, and then from IR adipocytes. This phenomenon was previously observed in the hepatocyte cell line HepG2 and in C2C12 myotubes; thus, Kranendonk and collaborators found that EVs from subcutaneous and visceral adipose tissue of some individuals inhibit Akt phosphorylation on the hepatic and muscle cells [15]. Additionally, others have described that adipose tissue exosome-like vesicles mediate the activation of macrophage- induced insulin resistance [11]; and a pro-inflammatory effect on macrophages was described for AT derived exosomes especially those derived from visceral fat [10]. Under this context, we also examined the role of pathological EVs isolated from the established cell models showing parallel results that confirm the capacity of hypertrophied adipocytes to induce macrophage inflammation through EVs. We assayed physiological, half, and a double dose of vesicles showing that it was necessary to apply double dose of OLEIC EVs to induce inflammation; on the contrary, a physiological dose of PALM EVs was enough to induce a significant TNFα and IL-6 expression on macrophages. On the other hand, we observed that EVs from the IR cell model HGHI did not exert an inflammation effect on macrophages, suggesting the necessity of lipid hypertrophy in this process as previously described [58]. 

The limitations of this study reside in the murine cell model itself that may not represent real obese hypertrophied or IR adipocytes; moreover, we are not considering the interaction with other cell types normally present in the adipose tissue such as those from the stromal vascular fraction, or the immune cells that characteristically invade obese adipose tissue. However, many of the described proteins on the EVs isolated from the established pathological cell models were recently identified by us in vesicles, shed by whole human obese adipose tissue and validating the results herein described. On the other hand, the methodology limitations have to be taken into account, which include the arduous vesicles isolating protocol and the mass spectrometry itself; for certain proteins, although detected by MS in the qualitative analysis, their quantity changes were not detected due to the restrictive statistical parameters, however, they were detected by immunoblot instead (mimecan). 

All together, we describe for the first time the EVs protein cargo of adipocytes subjected to different metabolic insults and its quantitative variation. The role of intracellular proteins on EVs is still not known, neither their possible function once they reach a target cell; in the meantime, the knowledge about these proteins may be very valuable as disease biomarkers since we have proved that EVs content varies depending on the pathological condition, being a representation of what is happening inside the cell of origin. Thus, we suggest several potential candidate biomarkers that will be validated in the near future. On the other hand, the functional assays in the present manuscript show that IR and hypertrophied adipocytes release Trojan horse like-vesicles able to induce metabolic alterations on healthy cells, probably exacerbating the disease once established which deserves further investigation.

## 4. Materials and Methods 

### 4.1. Cell Culture and Models

The murine MSC (mesenchimal stem cell line) C3H10T1/2 cells were obtained from Eduardo Domínguez Medina, and it was cultured at 37 °C under 5 % CO2 in DMEM (4.5 g/L glucose, LONZA) containing 10% fetal bovine serum (SIGMA-ALDRICH, MO, USA), 100 U/mL of penicillin, and 100 μg/mL of streptomycin (SIGMA-ALDRICH, MO, USA) until differentiation to adipocytes as previously described [59]. In brief, differentiation was induced after confluence with induction medium (basic medium supplemented with 1 µM dexamethasone, 0.5 mM isobutylmethylxanthine, 1 µM rosiglitazone (SIGMA-ALDRICH, MO, USA), and 5 µg/mL insulin (Actrapid, NovoNordisk)). Two days after induction, the medium was changed to basic medium with insulin. Accumulation of cytoplasmatic triglyceride in these cells was detected by staining with Oil Red O (SIGMA-ALDRICH, MO, USA). 

High glucose and high insulin (HGHI) insulin resistance model was established as described previously [60]. Briefly, cells were differentiated until day 6, and then were washed three times with PBS and incubated during 2 h in low glucose (1 g/L) cell culture medium. Cells were then cultured in high glucose (4.5 g/L) and high insulin (100 nM), or with palmitate [500 µM in 2% fatty acids-free BSA (SIGMA) or oleic acid (1 mM, conjugated in 0,5 mM fatty acids-free BSA (SIGMA)] in serum-free medium for 24 in oleic and HGHI, and 18h in palmitate. For vesicle isolation, this treatment was prolonged up to 48 h. For P-Akt/Akt pathway analysis cells were then washed three times in PBS and stimulated with insulin (100 nM) for 10 min.

The murine macrophages Raw 264.7 cells were cultured in DMEM (4.5 g/L glucose, LONZA) containing 10 % fetal bovine serum (SIGMA-ALDRICH, MO, USA), 100 U/mL of penicillin, and 100 μg/mL of streptomycin (SIGMA-ALDRICH, MO, USA) at 37 °C under 5 % CO_2_.

### 4.2. Immunobloting

Protein extracts from cells were obtained as previously described [61]. Briefly, proteins samples were extracted by homogenization using a TissueLyser II (QIAGEN, Tokio, Japan) in cold RIPA buffer (200 mM Tris/HCl (pH 7.4), 130 mM NaCl, 10% (*v*/*v*) glycerol, 0.1% (*v*/*v*) SDS, 1% (*v*/*v*) Triton X-100, 10 mM MgCl_2_) with anti-proteases and anti-phosphatases (Sigma-Aldrich; St Louis, MO, USA). Twenty-five μg cell lysates and the VEs obtained from the same amount of cells per plate from at least three independent experiments were separated in 10% SDS-PAGE gels and electroblotted onto nitrocellulose membranes as previously described [13]. Primary anti-PAkt, and anti-Akt were purchased from Cell Signaling Technology (MA, USA); anti- Alix, anti-Ceruloplasmin, anti-TSG101, anti-CD81, anti-Syntenin, and anti-Mimecan from Sta. Cruz Biotechnology (CA, USA); anti-TGFBeta Ig-h3 (NMP2-67186; dilution 1:500) from Novus Biologicals (NovusBio, CO, USA) and anti-Perilipin-1 from Abcam. Data were expressed as percentages corrected towards GADPH (arbitrary units) in Western blots with mean ± SEM. Data analyses were conducted using GraphPad Prism 6 software.

### 4.3. Cells Secretome Collection and Vesicle Isolation

Control adipocytes and those exposed to palmitate, oleic acid, and HGHI were washed twice with PBS and then incubated during 48 h in serum-free DMEM. The resulting secretome was collected and centrifuged at 1800 rpm for 5 min to be then filtered in a 0.22 µm filter to remove contaminating cell debris. This secretome was kept at –80 °C until ultracentrifugation. Following previous reports [62,63], secretomes were centrifuged in a Beckman Coulter OptimaTM L-100XP at 10.000 g at 4 °C for 20 min, followed by 100.000 g ultracentrifugation at 4 °C for 90 min with a Type SW 40Ti rotor, acceleration and deceleration brake profile 9, to pellet vesicles. The supernatant was carefully removed, and vesicle-containing pellets were resuspended in ice-cold PBS. A second round of ultracentrifugation (100,000 g at 4 °C for 90 min) was performed. The final pellet with the isolated vesicles was resuspended in ice-cold PBS or in RIPA sample buffer according to the analysis (Figure 1A).

### 4.4. Immunogold and Scanning Electron Microscopy

Immunolabeling of exosomes with antibodies was performed according to Thèry and collaborators [64]. Purified EVs were fixed in 2 % paraformaldehyde, and 5 μL were deposited on Formvarcarbon coated electron microscopy grids, and were left to air-dry for 1 h. Grids were then washed three times with PBS, followed by a 10-min incubation with 0.1 M glycine. Next, grids were incubated with 1 % BSA in PBS for 10 min at RT. Primary antibody (CD9; 1:10) was diluted in PBS + 1 % BSA and a 5 μL-drop was deposited over the grid for an O/N incubation at 4 °C. Excess antibody was eliminated by washing the grids with PBS + 1 % BSA. Secondary gold-labeled antibody was added for 1h at RT. Excess secondary antibody was also eliminated by washing the grids six times with PBS + 1% BSA. Grids were then incubated with 1 % glutaraldehyde for 5 min, followed by several washes with distilled water. Grids were incubated with uranyl-oxalate pH 7 for 5 min and then contrasted in a mixture of 4% uranyl acetate and 2 % methyl cellulose, for 10 min on ice. Grids were air dried and visualized using the JEOL JEM 1010 transmission electron microscope.

### 4.5. Nanoparticle Tracking Analysis (NTA)

Vesicle size and concentration distribution was carried out using Nanoparticle Tracking Analysis (NTA) in Malvern NanoSight NS300 equipment (v3.3 Malvern Panaytical, Ltd., UK) according to manufacturer’s instructions. Briefly, samples were vortexed and diluted to a final dilution of 1:100 in milliQ H_2_O. Blank-filtered H2O was run as a negative control. Each sample analysis was conducted during 60 s and measured five times using Nanosight automatic analysis settings. The detection threshold was set to level 4 and camera level to 14.

### 4.6. Mass Spectrometric Data Dependent Acquisition Qualitative Analysis (DDA) and Protein Quantification by Data Independent Acquisition (DIA) by Sequential Window Acquisition of All Theoretical Mass Spectra (SWATH) 

In order to make the qualitative and quantitative identification of EVs cargo proteins, we processed the vesicles obtained from the same number (9 confluent p60mm plates) of differentiated and undifferentiated C3H10T1/2 cells, and from the HGHI, palmitate and oleic cell models of 4 independent experiments (3 replicates for each experiment *n* = 12). Digested peptides were separated by reverse phase chromatography. The gradient was created using a microfluid chromatography system (Eksigent Technologies nanoLC 400, SCIEX, Foster City, CA, USA) coupled to a Triple TOF 6600 high-speed mass spectrometer (SCIEX) with a microflow source as described in the Appendix A methods using a data dependent acquisition (DDA) method. For this analysis, only those proteins with FDR < 1% (99% confidence in the protein) were selected. For relative quantification by the SWATH DIA method, we first built a spectral library that groups each condition group into a pool. Then, equal amounts of each sample type (*n* = 12) were run in the TripleTOF 6600 using a SWATH-MS acquisition method, as described in detail in the Appendix A methods.

### 4.7. Protein Functional Analysis

Functional analysis was performed by FunRich open access software (Functional Enrichment analysis tool version 3.1.3) for functional enrichment and interaction network analysis (http://funrich.org/index.html) [65], Panther on line Classification System (http://www.pantherdb.org) [66], and Reactome on line pathway database (https://reactome.org) [67].

### 4.8. Functional Vesicle Assays 

The extracellular vesicles isolated from the HGHI, palmitate, oleic acid models, and from mature differentiated C3H10T1/2 cells were added over differentiated C3H10T1/2 adipocytes and over murine macrophages RAW 264.7 in culture. To treat cells with the most physiological vesicle concentration possible, we established a 1:1 ratio; vesicles from a determined number of cells in culture were added to the same number of target cells.

To observe the possible changes in the insulin signaling pathway, C3H10T1/2 cells were differentiated as previously described. On day 6 of differentiation, after three washes with PBS followed by 2 h incubation with low glucose culture medium (1 g/L), cells were cultured with VEs isolated from the HGHI, palmitate, and oleic cell models during 24 h. In the case of combined treatments, adipocytes at day 6 of differentiation were cultured first in the presence of EVs from palmitate or oleic treated cells for 24 h, and then EVs of the HGHI cell model were added and incubated for another 24 h. After this time, three PBS washes were made before stimulation with insulin (100 nM) for 10 min. Four independent experiments were performed to subsequently analyze the P-Akt / Akt pathway.

To assay differentiation, 3000 C3H10T1/2 cells were differentiated, as previously described, in E-plates, and differentiation was measured by xCELLigence real-time impedance-based biosensor system (RTCA, ACEA Biosciences, San Diego, CA, USA). These cells were incubated with VEs from the three cell models (HGHI, palmitate and oleic acid) and cell index recorded at days 0, 2, 4, and 6 of differentiation (*n* = 3 independent experiments, 4 replicates each experiment). Cell culture medium with the same amount of EVs was refreshed every two days, and PBS vehicle was used as a control. 

For inflammation experiments, murine macrophages (Raw 264.7) were cultured in the presence of EVs isolated from differentiated control C3H10T1/2 adipocytes, and HGHI, palmitate and oleic cell models at a physiological concentration (1 EVs) and also at half (0.5 EVs) and double (2 EVs) doses for 24 h. Subsequently, RNA was extracted for further real time PCR analysis. The negative control was the PBS vehicle, and treatment with 1ug/mL of LPS (Lipopolysaccharides from Escherichia coli O111:B4, Sigma) as a positive control of the state of inflammation (*n* = 6 independent experiments were performed).

### 4.9. Quantitative Real-Time PCR

RNA was isolated using TRIzol protocol (TRI Reagent, Sigma^®^ #T9424), in which Chloroform (Sigma^®^ #C2432) is as denaturalization agent for the cells and Isopropanol (Sigma^®^ #650447) is the RNA precipitating agent. The precipitate was diluted in DEPC water (Ambion^®^ #AM9920) containing 1% SUPERaseIn RNase Inhibitor (Thermo Fisher^®^ #AM2694). One μg of RNA was treated with DNase I (Invitrogen^®^ #18068015) prior to their reverse transcription by SuperScript^®^ III First-Strand Synthesis System kit (Invitrogen^®^, #18080051). The cDNA obtained was employed for the gene expression analysis, which was performed using FastStart Universal SYBR GREEN Master (ROX) (Roche^®^ #04913914001) in a Mx3005P system (Agilent Technologies^®^). The PCR amplification conditions were a denaturating step with 30 s at 95 °C; 45 cycles of 30 s at 95 °C, 1 min at 60 °C, and 30 s at 72 °C; followed by a cooling step. Each sample was run in triplicate in each experiment. Primers used: IL6 (forward, 5′-CCCAACAGACCTGTCTATACCA-3′; reverse, 5′-CAGAATTGCCATTGCACAAC-3′), TNFα (forward, 5′-CCACCACGCTCTTCTGTCTA -3′; reverse, 5′- AGGGTCTGGGCCATAGAACT-3′) and HPRT as housekeeping (forward, 5′-CAATGCAAACTTTGCTTTCCC -3′; reverse, 5′-TCCTTTTCACCAGCAAGCTTG -3′).

### 4.10. Statistical Analysis

To achieve the global qualitative proteome analysis data files were processed using ProteinPilotTM 5.0.1 software (SCIEX) which uses the algorithm ParagonTM for database search and ProgroupTM for data grouping. Data were searched using a Human specific Uniprot database (released January 2018). False discovery rate was performed using a non-lineal fitting method displaying only those results that reported a 1% Global false discovery rate or better [68]. 

To make the SWATH quantitative analysis, the targeted data extraction of the fragment ion chromatogram traces was performed by PeakView (version 2.2) using the SWATH Acquisition MicroApp (version 2.0). For the PCA and cluster analysis the R 3.5.3 version and “base”, stats”, “gplots”, “Hmisc”, and “car” packages have been used after normalization of raw data. Principal Component Analysis (PCA) was applied, taking into account correlation matrix due to its pair wise two-sided p-values are 0 for the entire matrix, and therefore all of them are statistically significant, which could explain a good success in the application of PCA. Therefore, 87.70% of the total variability of the data is explained with the two first principal components. For cluster analysis, unpaired Student’s t-Test for means comparison between each and differentiated samples was previously applied, and used Euclidean distance, suitable for quantitative variables and complete linkage as cluster criteria. For differentially expressed protein selection, a fold change ≥ 1.5 and a *p*-value ≤ 0.05 were selected. 

The statistical significance among multiple groups was analyzed by one-way Anova-Kruskall Wallis test followed by Dunn’s multiple comparison test and the comparison of the results between two groups was done by Mann–Whitney U test for comparison of results between two groups using in GraphPad Prism 6 software. *p* ≤ 0.05 was considered statistically significant.

## 5. Conclusions

In this manuscript, we describe for the first time the extracellular vesicles’ (EVs) protein cargo from adipocytes subjected to different metabolic insults and its quantitative (SWATH) variation by mass spectrometry analysis.

Protein content of EVs shed by mature adipocytes was modified and altered by lipid hypertrophy and insulin resistance (IR).

EVs secreted by hypertrophied and IR adipocytes carry obesity and comorbidities-related proteins; a selection of those is suggested as obesity EV-biomarkers.

EVs isolated from lipid hypertrophied adipocytes are characterized by the elevated presence of ceruloplasmin, mimecan and perilipin 1 adipokines.

EVs isolated from IR adipocytes showed an exclusively striking presence of the IR related protein TFGBI.

EVs shed by hypertrophied and IR adipocytes promote the pathology of the cell of origin in normal healthy adipocytes; thus, upon interaction with target cells, pathological vesicles induce differentiation and hypertrophy, and promote insulin resistance by inhibiting insulin pathway (decrease of P-Akt). 

EVs liberated by lipid atrophied adipocytes promote macrophage inflammation.

Thus, we suggest that obese adipocytes release Trojan horse-like vesicles able to induce metabolic alterations on healthy cells, probably exacerbating the disease once established, deserving further investigation.

## Figures and Tables

**Figure 1 ijms-21-02252-f001:**
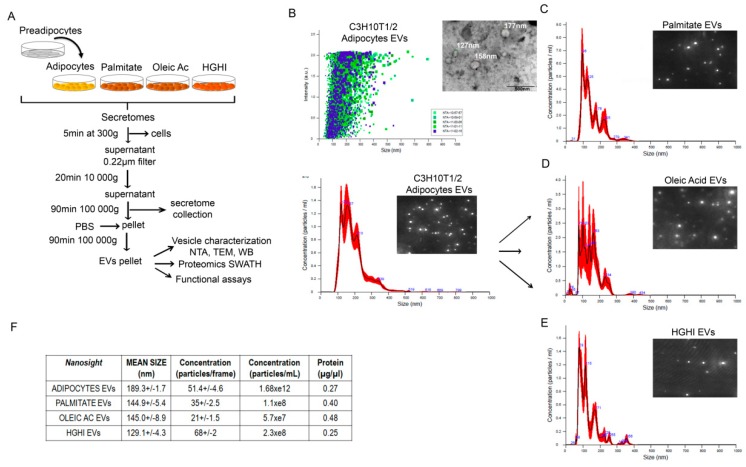
Adipocytes secrete extracellular vesicles whose size and concentration varies with metabolic insults. Schematic representation of vesicle isolation protocol by differential centrifugation and further analysis from secretomes obtained from control C3H10T1/2 adipocytes and after metabolic insults with palmitate (500 µM, 18 h), oleic acid (1 mM, 24 h) and high glucose (4.5 g/L)/high insulin (5 µg/mL, 24 h) treatment (HGHI) (**A**). Nanoshight analysis of isolated vesicles showing size distribution (nm) and particle dispersion (**B**–**E**); a representative TEM image of vesicles isolated from differentiated adipocytes is also shown (**B**). Mean size (nm) and estimated concentration is shown as particles/frame, particles/mL of secretome, and protein concentration (µg/µL) obtained after centrifuging secretomes from 18 cell plates of 35 mm (**F**). NTA: nanoparticle tracking analysis; TEM: transmission electron microscope; WB: western blot.

**Figure 2 ijms-21-02252-f002:**
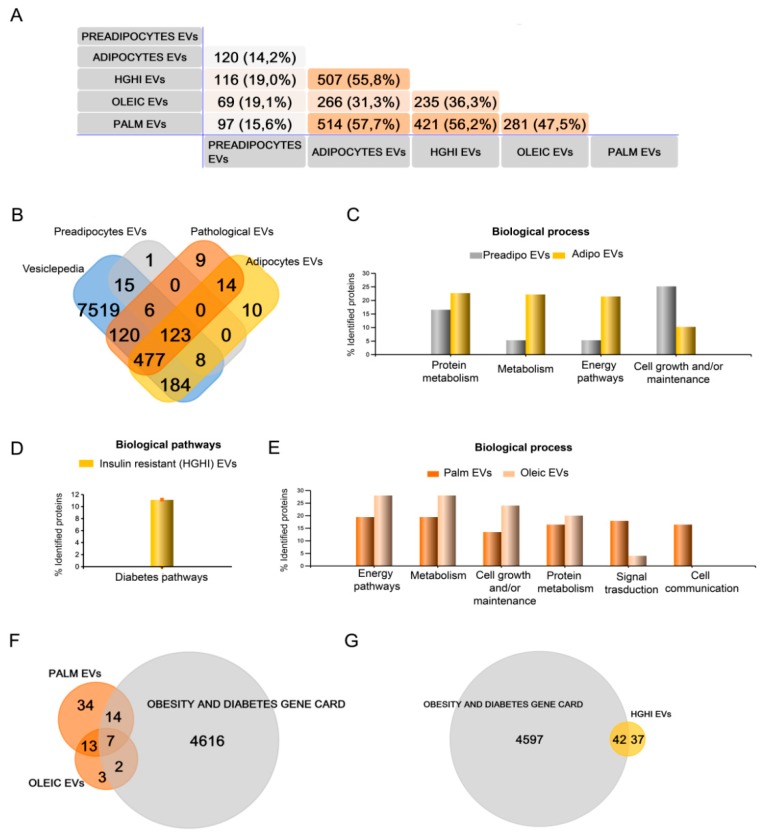
Qualitative (DDA) mass spectrometry analysis shows that extracellular vesicles (EVs) protein cargo varies with lipid hypertrophy and insulin resistance. EVs isolated from non-differentiated and differentiated adipocytes as control, and also from those treated with palmitate (500 µM, 18 h), oleic acid (1 mM, 48 h), and high glucose (4.5 g/L)/high insulin (5 µg/mL, 48 h) treatment (HGHI) were analyzed by mass spectrometry. A descriptive Venn diagram showing the overall number of proteins identified with an FDR <1% (99% protein confidence) in vesicles isolated from preadipocytes, control differentiated adipocytes, and those from the three pathological cell models (HGHI, palmitate and oleic acid hypertrophy) (*n* = 4 independent experiments considering only those proteins present in 3 out 4 samples) (**A**). All identified proteins were also compared to the Vesiclepedia database (**B**). Funrich functional analysis classification of identified proteins by biological process and pathways is shown (percentages per annotated genes in each class; number of genes in background) for vesicles isolated of pre- and mature adipocytes (**C**); of HGHI (**D**), and of palmitate and oleic acid treated cells (**E**). Venn diagrams comparing the Obesity and Diabetes Gene card database to those proteins exclusively identified (not present in EVs shed by control differentiated cells) in EVs from lipid hypertrophied (**F**) and HGHI (**G**) treated adipocytes.

**Figure 3 ijms-21-02252-f003:**
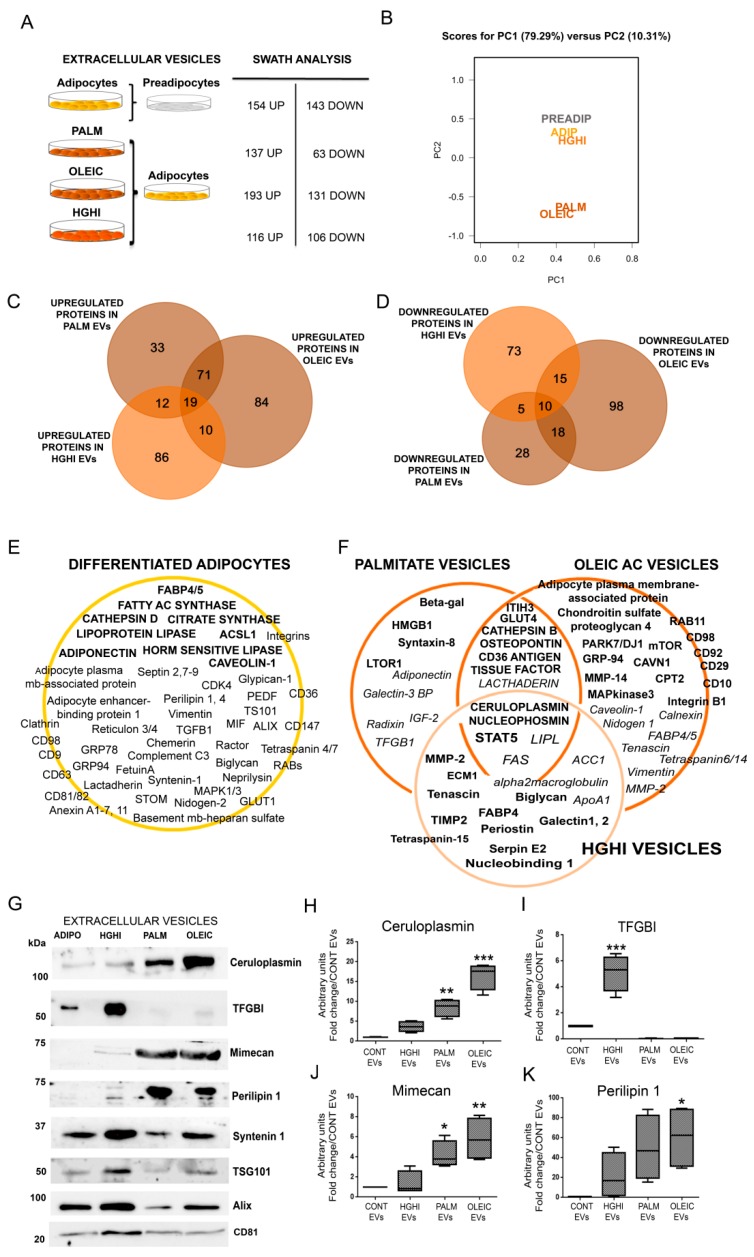
Quantitative Sequential Window Acquisition of All Theoretical Mass Spectra (SWATH) analysis of EVs shed by lipid hypertrophied and insulin resistant adipocytes identifies obesity-related proteins and potential biomarkers. Differences on protein cargo of adipocytes vs. preadipocytes, and also in those vesicles shed by palmitate, oleic, and HGHI treated cells compared to control adipocytes by comparative analysis of EVs by Sequential Window Acquisition of all theoretical fragment-ion spectra is indicated (**A**). Principal component analysis (PCA) analysis of transformed SWATH areas for the quantitative comparison of all samples (*n* = 4 independent experiments; *n* = 3 technical replicates for each independent experiment) (**B**). Venn diagrams comparing those proteins upregulated (**C**) and downregulated (**D**) in EVs from the three pathologic cell models is shown. A representative proteome map of EVs secreted from control adipocytes compared to non-differentiated cells is represented showing representative upregulated proteins in bold (**E**). A proteome map highlighting selected proteins up (bold) and down (italic) regulated in EVs isolated from lipid hypertrophied (palmitate/oleic acid) and insulin resistant (HGHI) adipocytes (**F**). Representative images of immunoblots and band quantification in box plot graphs for cereluplasmin, transforming growth factor-beta-induced protein ig-h3 (TFGBI), mimecan, perilipin 1, syntenin 1, TSG101, Alix, and CD81 in independent isolated vesicles from all the cell models is shown (*n* = 4 independent lysates) (**G**–**K**); differences were assayed by one-way Anova-Kruskall Wallis test followed by Dunn’s multiple comparison test (*p* ≤ 0.05 was considered statistically significant: * *p* < 0.05, ** *p* < 0.01, and *** *p* < 0.001).

**Figure 4 ijms-21-02252-f004:**
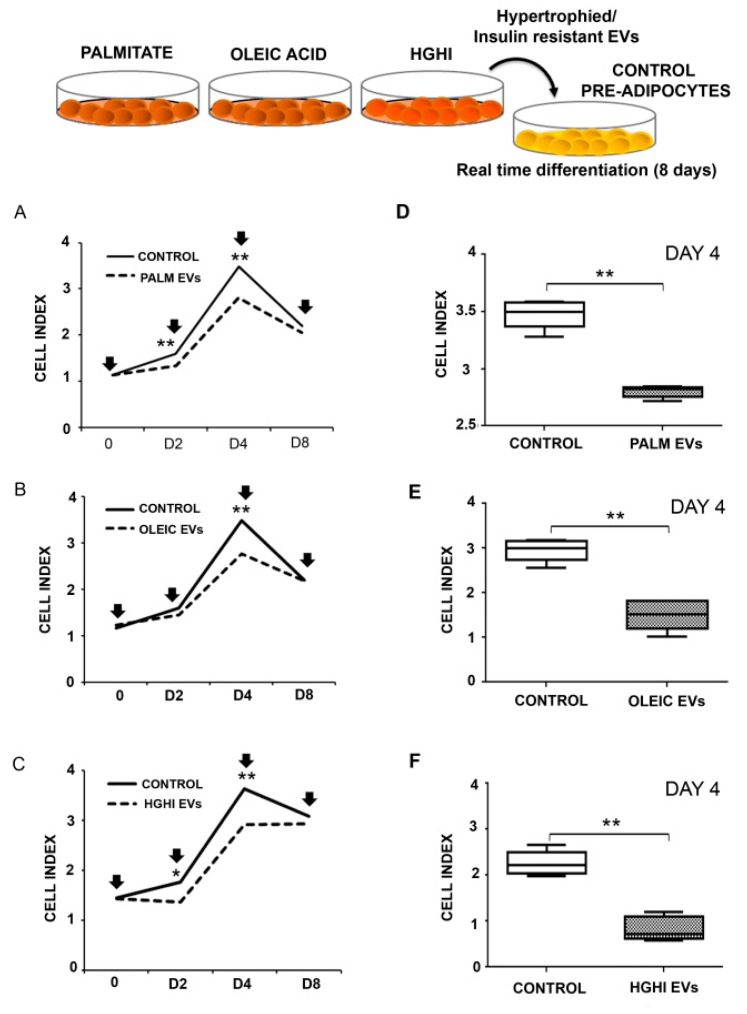
EVs secreted by lipid hypertrophied and insulin resistant adipocytes stimulate neighbored cells atrophy. Real time differentiation of C3H10T1/2 preadipocytes incubated with pathological vesicles (**A**: Palmitate EVs; **B**: Oleic Acid EVs; and **C**: HGHI EVs) isolated from an equivalent amount of cells (vesicles isolated from the same amount of treated cells) is shown. Cell index graphs are shown from day 0 to 8 of differentiation (3 independent experiments with *n* = 4 replicates/experiment; vesicles were refreshed every 48 h: days 0, 2, 4, 6, and 8 of differentiation indicated with a black arrow). Box diagrams showing the cell index at day 4 of differentiation for those adipocytes treated with pathological EVs is shown (**D**–**F**). Comparisons between two groups were performed by Mann-Whitney U test; *p* ≤ 0.05 was considered statistically significant: * *p* < 0.05 and ** *p* < 0.01.

**Figure 5 ijms-21-02252-f005:**
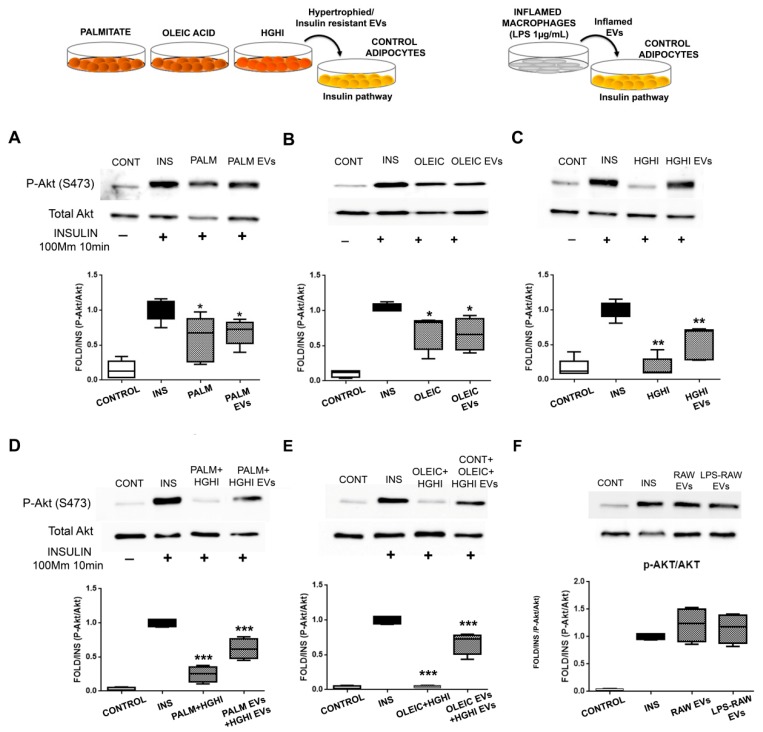
EVs shed by pathological adipocytes exert insulin resistance in adipocytes. The effect of EVs from lipid hypertrophied (palmitate/oleic), insulin resistant (HGHI) adipocytes (**A**–**E**), and EVs from inflamed macrophages (**F**) on insulin pathway assayed on control adipocytes is shown. Representative images and densitometry of bands expressed toward insulin (10 mM, 10 min stimuli) of P-Akt and total Akt immunoblots ((*n* = 4 independent lysates) are shown; differences were assayed by one-way Anova-Kruskall Wallis test followed by Dunn´s multiple comparison test (*p* ≤ 0.05 was considered statistically significant: * *p* < 0.05, ** *p* < 0.01, and *** *p* < 0.001). CONT: differentiated adipocytes without insulin stimulation; INS: differentiated adipocytes stimulated with 10 mM insulin for 10 min; PALM/OLEIC/HGHI: palmitic/oleic-hypertrophied/HGHI treated adipocytes stimulated with 100 mM insulin for 10min; PALM/OLEIC/HGHI EVs: differentiated adipocytes treated (24 h) with EVs shed by palmitate or oleic hypertrophied, or by HGHI adipocytes and then stimulated with 100 mM insulin for 10 min; PALM + HGHI EVs: control adipocytes treated (24 h) with EVs secreted by palmitic hypertrophied cells followed by treatment with HGHI EVs (another 24 h), and then stimulated with 100 mM insulin for 10min; OLEIC + HGHI EVs: control adipocytes treated (24 h) with EVs secreted by oleic hyperthrophied cells followed by HGHI EVs treatment (another 24 h), and then stimulated with 100 mM insulin for 10 min; RAW EVs: control adipocytes treated (24 h) with EVs secreted by control non-inflamed macrophages followed by 100 mM insulin stimulation for 10 min; LPS-RAW EVs: control adipocytes treated (24 h) with EVs secreted by LPS (1 µg/mL, 24 h) inflamed macrophages followed by 100 mM insulin stimulation for 10 min.

**Figure 6 ijms-21-02252-f006:**
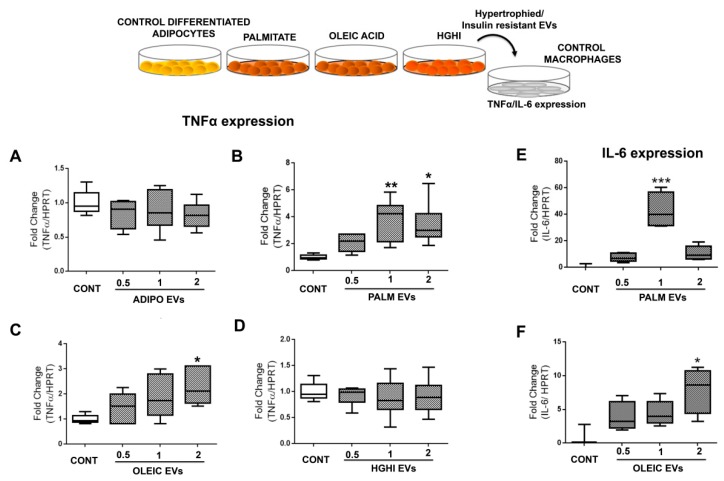
EVs shed by hypertrophied adipocytes exert inflammation in macrophages. The effect of vesicles from lipid hypertrophied and HGHI adipocytes at different concentrations (physiological (1), half (0.5) and double doses (2)) over control non-inflamed macrophages was assayed by real time PCR expression of TNFα (**A**–**D**) and IL-6 (**E**–**F**). Box graphs are shown as fold change towards control adipocytes without any stimulus corrected by HPRT (at least 4 independent experiments; *n* = 3 replicates/experiment). Differences were assayed by one-way Anova-Kruskall Wallis test followed by Dunn´s multiple comparison test (*p* ≤ 0.05 was considered statistically significant: * *p* < 0.05, ** *p* < 0.01, and *** *p* < 0.001).

**Table 1 ijms-21-02252-t001:** List of suggested potential EV biomarkers for pathological adipocytes. Selected proteins identified in the proteomics analysis are suggested as EVs biomarkers based in the bibliography and in previous analysis performed by us in EVs from human obese adipose tissue (Camino et al., under review). Up-arrow: up regulated; down-arrow: down regulated in the SWATH analysis. IR: insulin resistance; MS: metabolic syndrome; AT: adipose tissue.

EV Potential Biomarker	Protein ID	Palm EVs	Oleic EVs	HGHI EVs	Human Obese AT EVs	Pathology	Reference
72kDa type IV collagenase (MMP-2)	MMP2_MOUSE			X↑	VISCERAL OBESE AT	IR + obesity	Derosa et al., 2008; Mazor et al., 2018 [20]; Camino et al., 2020 under review [19]
Cathepsin B	CATB_MOUSE	X	X			obesity, inflammation, MS	Araujo et al., 2018 [21]
CD36	CD36_MOUSE	X	X			inflammation + IR	Kennedy et al., 2011 [22]
Ceruloplasmin	CERU_MOUSE	X ↑	X↑	X	VISCERAL OBESE AT	obesity	Arner et al., 2014 [23]; Kim et al., 2011 [24]
FABP-4	FABP4_MOUSE			X	VISCERAL OBESE AT	obesity + MS	Xu et al., 2006 [25]; Camino et al., 2020 under review [19]
ITIH3	ITIH3_MOUSE	X	X			inflammation	Fries et al., 2003 [26]
Mimecan	MIME_MOUSE	X ↑	X↑	X	VISCERAL OBESE AT	adipokine	Cao et al., 2015 [27]
MMP-14	MMP14_MOUSE		X			IR	Li et al., 2020 [28]
Osteopontin	OSTP_MOUSE	X	X			inflammation + IR	Zeyda et al., 2011 [29]
PARK7/DJ-1	PARK7_MOUSE	X	X↑			obesity-inflammation	Kim et al., 2014 [30]
Perilipin 1	PLIN1_MOUSE	X↑	X↑	X		AT Ev Biomarker	Eguchi et al. 2016 [31]
Syntasin-8	STX8_MOUSE	X				VAT IR + inflammation	Lancha et al., 2015 [28]
TFGBI	BGH3_MOUSE			X↑↑	VISCERAL OBESE AT	diabetes	Camino et al.,2020 under review [19]; Roca-Rivada et al., 2015 [32]; Fain et al., 2005 [33]
Tissue factor	TF_MOUSE	X	X			Obesity + inflammation	Samad et al., 1998 [34]; Badeanlou et al., 2011 [20]

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
