# Peer review of "Vesicles Shed by Pathological Murine Adipocytes Spread Pathology: Characterization and Functional Role of Insulin Resistant/Hypertrophied Adiposomes"

_ijms, 2020, doi:10.3390/ijms21062252_

Round 1

Reviewer 1 Report

The manuscript is very extensive in terms of both quantity and quality. The results presented by the authors are meticulous. 
"Instructions for authors" of International Journal of Molecular Sciences recommend: "Conclusions: This section is not mandatory, but can be added to the manuscript if the discussion is unusually long or complex". The volume of the reviewed manuscript may make it difficult for readers to find the most important results that could be cited and help other authors in their research work. In view of the above, it seems necessary to attach the conclusion section - optimally organized in the form of several key points.

The evaluation of similarity to the second manuscript prepared by the authors that is currently under review seems to be very difficult since the paper has not been published and is unvailable. The authors informed that "we compared identified proteins to our previous study analyzing EVs from human obese whole visceral and subcutaneous adipose tissues (Camino et al., under review) finding several common proteins (Supp tables 3-5)." This is especially important due to the fact that obesity and insulin resistance are related to each other in many aspects.

Author Response

We thank the reviewer for the comments about our manuscript. We are pleased to know about the quality of our work, and we are happy to follow the reviewer recommendation in relation to the conclusions section. We agree that adding key points highlighting the main results of the research will be very helpful for other authors. Please see that we have added these in a Conclusions section.

Moreover, we agree with the reviewer about the comparison with our previous work may seem difficult; however, we believe that this comparative analysis is necessary and adds importance to the findings in this manuscript were we show obesity in vitro cell models. We were trying to speed up the reviewing process of our previous work on EVs from human obese adipose tissue explants, submitted to Translational Research, to hopefully provide a definitive reference. Reviewers asked us to add some, not very substantial experiments, for a second round of revision. However, we are actually totally stuck due to the current situation here in Spain due to the COVID19 crisis since we are confined at home. We will be happy to reference our work as you and the Editor decide because we really believe that this comparison adds a great value to the current work. We are actually performing an enquire to the Editors to decide how to proceed.

Reviewer 2 Report

The authors isolated and characterized vesicles shed by a murine model of adipocyte differentiation during lipid atrophy and insulin resistance. Differences between normal and hypertrophied/insulin- resistant adipocyte-vesicle loads were identified and characterized. The authors evaluated the functional effect of vesicles isolated from pathological hypertrophied and insulin- resistant adipocytes on healthy adipocytes and macrophages. The authors concluded that pathological adipocytes release vesicles containing a representative protein cargo of the cell of origin that are able to induce metabolic alterations on healthy cells probably exacerbating the disease once established. The study was original, and the data were solid. The reviewer has one suggestion. In the discussion section, it would be useful to show the EVs protein load of mesenchymal stem cells (MSCs). It would be correct to compare the protein load of EVs MSCs with the protein load of EVs adipocytes. It is known that MSCs are contained in adipose tissue and can differentiate into adipocytes. On the other hand, MSCS have anti -inflammatory activity. It is necessary to give an explanation for this.

Author Response

We thank the reviewer for the positive comments on our manuscript and the suggestion. It is true that have we focused our analysis on those proteins present on the EVs from IR and hypertrophied adipocytes compared to mature adipocytes paying less attention to the non differentiated adipocytes, which are indeed contained within the stromal vascular fraction of adipose tissue. We appreciate this suggestion since we totally agree that it may offer valuable data. Please see that the protein content of EVs shed from preadipocytes and its comparison to those present in mature adipocytes are listed in the Suppl. Table 1; moreover, the quantitative analysis comparing the protein content among those vesicles, are listed in Supp. Table 6. Thus, as recommended by the reviewer, we have added the discussion of these results in the discussion section.

Round 2

Reviewer 1 Report

Dear Authors,

Thank you for attempting to address my concerns.

I accept the manuscript in the present form.